# The Short- and Long-Term Anticipation of Prostate Cancer Incidence in Korea: Based on Social Aging Trends and Prostate-Specific Antigen Testing Rate during the Last Decade

**DOI:** 10.3390/cancers16030503

**Published:** 2024-01-24

**Authors:** Jong Hyun Pyun, Young Hwii Ko, Sang Won Kim, Nak-Hoon Son

**Affiliations:** 1Department of Urology, Kangbuk Samsung Hospital, Sungkyunkwan University School of Medicine, Seoul 03181, Republic of Korea; docpjh79@hanmail.net; 2Department of Urology, College of Medicine, Yeungnam University, Daegu 42114, Republic of Korea; 3Medical Research Center, College of Medicine, Yeungnam University, Daegu 42415, Republic of Korea; kimsw3767@ynu.ac.kr; 4Department of Statistics, Keimyung University, Daegu 42601, Republic of Korea; nhson@ms.kmu.ac.kr

**Keywords:** incidence, prostate cancer, screening

## Abstract

**Simple Summary:**

This study aimed to anticipate the short- and long-term PCa incidence projected from the PSA testing rate (PSAr) trend and social aging during the last decade. We found that the incidence of PCa increased fourfold (4533 in 2006 to 16,815 in 2020), with each age subgroup accounting for 7.9% (50s), 31.4% (60s), 43.0% (70s), and 17.1% (over 80s) of cases in 2020. According to the projection of this study, the incidence of PCa will increase twofold by 2034 compared to 2020. If national screening were only carried out in the 60s and 70s age groups, a higher detection of almost threefold would be expected by 2040.

**Abstract:**

The current incidence of prostate-specific antigen (PSA) testing, which plays a crucial role in detecting prostate cancer (PCa) in an aged population, is low in Korea. Reflecting these epidemiologic characteristics, we estimated the short- and long-term incidences of PCa. A regression equation model was extracted based on two critical pieces of information: (1) the distribution of newly detected PCa cases in each age group of the 50s, 60s, 70s, and over 80s from a recent period (2006–2020), and (2) the PSA testing rate (PSAr) from the previous decade (2006–2016) for each age subgroup. The incidence increased fourfold (4533 in 2006 to 16,815 in 2020), with each age subgroup accounting for 7.9% (50s), 31.4% (60s), 43.0% (70s), and 17.1% (over 80s) of cases in 2020. PSAr increased by an average of 1.08% annually. If these trends are maintained, 28,822 new cases will be diagnosed in 2030 (expected PSAr: 14.4%) and 40,478 cases in 2040 (expected PSAr: 26.4%). If a public PSA screening were implemented for men only in their 60s (assuming a PSAr of 60% in the 60s) and 70s (assuming a PSAr of 80% in the 70s) in 2030, 37,503 cases in 2030 (expected PSAr: 23.1%) and 43,719 cases in 2040 (expected PSAr: 29.9%) would be estimated. According to the projection, the incidence of PCa will increase twofold by 2034 compared to 2020. If national screening were only conducted in the 60s and 70s, a higher detection of almost threefold would be expected by 2040.

## 1. Introduction

In 2020, prostate cancer (PCa) was the most frequently diagnosed cancer in men in over half (112 of 185) of all countries [1]. Although relatively lower PCa incidence and mortality rates have been observed in Asian men compared to those from Western areas, there is a considerable difference in incidence even within a similar geographical area [2]. For instance, PCa became the most common cancer among Japanese males, with 98,400 new cases being registered in 2015 [3]. In contrast, in South Korea, PCa was the fifth most common cancer with 10,212 patients in the same year. PCa remained the 10th most prevalent male cancer until the end of the 20th century [4].

Several factors, such as genetic susceptibility, lifestyle, and the degree of social aging, may contribute to these differences. However, given similar racial and cultural backgrounds, especially between Korea and Japan, international differences in diagnostic practices among Asian countries are more likely to generate most of the distinction [5]. Many of these differences may be explained by the opportunity of ordinary people of susceptible age for PCa to undergo prostate-specific antigen (PSA) testing [6,7].

In detecting PCa, PSA testing plays a pivotal role in the detection of a localized disease that accounts for over 90% of newly registered cases in developed countries because non-metastatic PCa does not manifest specific symptoms except vague male lower urinary tract symptoms (LUTSs) that originate more frequently from concomitant benign prostate enlargement. In patients with no metastases and no symptoms, the detection rate of PCa is known to be directly related to the PSA level, which is not disease-specific, but may raise a suspicion of PCa. About 18% of patients with PSA in the 4 to 10 ng/mL range have a subsequent positive biopsy [8]. PSA levels greater than 10 ng/mL confer a greater than 67% likelihood of biopsy-detectable PCa [9]. Although digital rectal examination (DRE) has been recommended for screening, its role is currently controversial [10], and multiparametric magnetic resonance imaging has been proposed to compensate for the lower sensitivity of ultrasonography. However, its validation for large-scale screening is still ongoing [11].

Therefore, the opportunity for exposure to PSA testing is practically linked with the chance to screen individuals suspected of having the disease. However, this test is not included in routine health screening programs in Korea, so the current PSA testing rate in men over 40 was only 7.3% nationwide in 2016 [12]. Meanwhile, the majority of the PCa cases were detected in men over 60 (91.6% in 2020). PCa incidence increased with age, reaching 7.9% in men in their 50s, 31.5% in those in their 60s, and 43% in men in their 70s [9]. Indeed, as PSA screening rates have gradually increased, the incidence of PCa in South Korea has also risen, totaling 18,697 in the most recent 2021 report [13].

Combining these unique epidemiologic characteristics of Korea, including the low social perception of PSA screening and the elderly population-driven incidence of PCa, this study aimed to anticipate the short- and long-term PCa incidence projected from the current trend of nationwide PSA testing and official demographics during the last decade.

## 2. Materials and Methods

### 2.1. Data sources for Nationwide PCa Incidence and PSA Incidence

For the annual statistics of PCa, the Korean Statistical Information Service (KOSIS) database provided by the National Statistical Office of Korea was used [14]. The KOSIS database is a national, population-based database of cancer incidence and is currently used to calculate the National Cancer Registry statistics, published annually [15]. The KOSIS database obtains detailed information on the national population in each age group of the 50s, 60s, 70s, and over 80s.

Korea’s National Health Insurance Service (NHIS) database was used for annual PSA testing rate statistics. The NHIS covers approximately 98% of the population and provides universal health coverage. The NHIS database includes diagnostic codes, procedures, and outcomes (deaths), and also offers sociodemographic information such as age, health insurance premiums, and residential area. Utilizing the procedure codes, including B5490, C4280, and C7428, males aged over 40 years who underwent PSA tests were identified. All personal identification numbers were encrypted before data processing, and the study procedures and ethical aspects were approved beforehand by our Institutional Review Board (approval number: YUMC 2019-11-012-002).

### 2.2. Statistics Estimating the PCa Population in the Future

To anticipate the newly developed PCa population, a regression equation model for the estimated incidence of PCa in the future was extracted based on two critical pieces of information from the past: (1) the number of PCa cases in each age group of the 50s, 60s, 70s, and over 80s from the last two decades (2006–2020), and (2) the PSA testing rate from the previous decade (2006–2016) in each age subgroup. The extracted equation then enabled the projection of the number of PCa cases in each age subgroup. Because the KOSIS also provides detailed population demographics for each age subgroup in the future for each year from 2022 to 2047, the total number of projected patients could be estimated.

The study endpoint was the estimated incidence of PCa in 2022, 2030, and 2040 if the trends of the last decade hold in Korea. The estimated PSA testing rate in each age group was supposed to have a maximum of 80%, even if the current trend was maintained. We also calculated the anticipated PCa incidence if PSA testing were implemented as a national cancer screening program around 2030 for men aged 60 and 70 only. This is a hypothetical situation, but assumes a collective implementation of minimal PSA screening in the two age groups with the highest incidence of PCa in the last decade of data.

## 3. Results

### 3.1. Current Trend of PCa and PSA Incidence in Each Age Subgroup

The incidence of PCa in Korea increased about fourfold (4533 in 2006 to 16,815 in 2020) during the study period. Within the same period, the average change in PSA testing rate increased by 1.08% each year. Based on this equation, the regression model obtained the reflected number of newly detected PCa cases in each age subgroup.
$${\hat{Y}}_{i}=\alpha +\beta_{1}\left({\mathrm{PSA}}_{i}\right)+\beta_{2}\left({\mathrm{EP}}_{i}\right)\;$$
* i = age group (40s and under, aged in the 50s, aged in the 60s, aged in the 70s, aged in the 80s); * PSA = PSA testing rate; * EP = estimated population.

### 3.2. Anticipated Future Trend of PCa in Each Age Subgroup

The estimated total number of PCa cases was 19,227 in 2022. If the current trend of the PSA testing rate was maintained, 28,822 new PCa cases are expected in 2030, when the PSA testing rate is expected to reach 14.4% (Figure 1). In 2040, the number would be 40,478 cases when the total PSA incidence is expected to be 26.4% (Table 1).

If public PSA screening were implemented for men in their 60s (assuming a PSA testing rate of 60% in the 60s) and 70s only (assuming a PSA testing rate of 80% in the 70s) in 2030, 37,503 cases in 2030 (expected PSA testing rate: 23.1%) and 43,719 cases in 2040 (expected PSA testing rate: 29.9%) are anticipated (Figure 2).

## 4. Discussion

With the accelerated trend of aging in society, the landscape in the incidence of malignant disease in Korea is changing rapidly because PCa affects the elderly population [16,17] and exhibits age-dependent increases in incidence rates [18,19,20]. It remained the 10th most prevalent male cancer at the end of the 20th century. However, in 2002, PCa became the fifth most common male cancer. In 2020, its incidence took third place among male cancers for the first time. During the last two decades, the crude incidence of PCa in Korea increased tenfold (2.9 in 2000 to 32.7 in 2020), and mortalities increased fourfold (1.2 in 2000 to 4.3 in 2020).

As the irreversible aging trend continues globally, the incidence of PCa is expected to increase rapidly. This prediction aligns with current research findings and the conclusions drawn from this study. A literature review identified three articles that projected the number of newly diagnosed cases of PCa among Korean males until April 2023 (Table 2). Anticipating the cancer incidence in 2022 based on the National Cancer Incidence database for the last two decades (1999–2019), Jung et al. reported that PCa is expected to become the most common cancer among Korean males (15.5%), followed by lung cancer (15.1%) and stomach cancer (12.2%). Their expected PCa incidence in 2022 was 22,391 cases [21]. However, utilizing the same data source of the National Cancer Incidence database, Park et al. anticipated that PCa would remain in second place among the most common male cancers by 2035, following lung cancer. The expected incidence of PCa in 2034 was 29,339 cases [22]. The International Agency for Research on Cancer (IARC) is an agency of the World Health Organization (WHO) responsible for managing and conducting the GLOBOCAN project that provides estimates of cancer incidence, mortality, and prevalence worldwide. Through the GLOBOCAN database, IARC predicts the future incidence for a given country from the current assessment in 2020 until 2040 [23]. Their estimated PCa incidence in Korea is 20,900 for 2030 and 26,828 for 2040.

As such, the authors note that despite using nearly identical data, there were significant differences in the expected number of patients. The traditional statistical methods of population and period do not account for PSA screening rates, which are the primary means of detecting PCa in the clinical setting. Therefore, this method makes it difficult to estimate accurately when PSA screening rates are low or change rapidly. In Korea, PCa ranked third in terms of incidence and prevalence in the 2020s and tenth in the early 2000s. Therefore, public medical awareness and social interest in PCa have only recently increased. As a result, the frequency of PSA and DRE tests, which play a crucial role in detecting PCa, is low, and they are not included in general national cancer screening or health checkup programs. As a result, PSA screening rates vary widely according to education, income, and place of residence. DRE tends to rarely be performed in clinical practice due to low reimbursement rates and the need for time-to-efficiency priority awards within a limited time in the outpatient office. With this in mind, the authors statistically derived an equation that can compensate for the fact that PSA screening rates and PCa detection rates have varied dramatically by age in Korea over the past decade and used the results of this equation to estimate future projections.

Although there are differences in magnitude, all three studies’ data, taken from different sources and methodologies, similarly suggest that the incidence of PCa in Korea will continue to increase for the foreseeable future. Another aspect these three studies have in common is that they use demographic methods without considering the clinical context of PCa at the time of detection, so they do not provide specific projections for the future. For instance, the awareness of PSA testing is currently low in Korea. Indeed, the awareness of PSA testing as a screening tool for PCa remained at 9.7% among those in their 40s or older in a 2019 survey [24]. Nevertheless, all three studies assume that this low social perception will be maintained. However, with the recent surge in interest in prostate disease among the aging population and the increased prescription of medications that provoke PSA testing, such as 5-alpha reductase inhibitors for prostate enlargement, social awareness of PSA testing is expected to continue to grow. From this study, it was found that the average change in PSA testing rates has increased by 1.08% each year from the data between 2006 (2.54%) and 2016 (7.27%) and is expected to accelerate, matching the trend of social aging.

In this background, the present study’s unique advantage is that this approach could explain and project an association between the PSA testing rate and the estimated number of PCa cases. Suppose the current increasing trend of the PSA testing rate is maintained. In that case, the number of newly diagnosed patients will double by 2034 (33,692 cases, compared with 16,815 from the official data in 2020). The PSA testing rate for the entire male population in 2034 was expected to reach 19.2%. However, this estimated figure is about half that of the current data from the US. A national survey from 2005 to 2015 showed that the proportion of American males over 50 who had undergone a PSA test was slightly lower than the maximum estimate of 43.1% in 2008. According to recent research, it has since remained above 30% (32.8% in 2013 and 33.8% in 2015 [25]). Following the first prohibitive recommendations for PSA screening issued by the USPSTF in 2008 for men over 75 [26,27], a clear diminishing trend was observed among men over 70 (from 51.1% in 2008 to 36.4% in 2015). However, among men aged 55 to 69, the mean rate of PSA testing has increased (from 49.8 to 55.8 tests per 100 person-years). This increasing trend in PSA was also observed among men aged 40 to 54 years and 70 to 89 years, outside of the USPSTF-recommended screening age group. The most recent data between 2016 and 2019 demonstrated a 12% increase in PSA testing nationwide [28]. In 2018, US Behavioral Risk Factor Surveillance System Data showed that the screening prevalence was 43% in veterans and 40% in nonveterans among those aged between 55 and 69 [29]. If the current rate of increase continues in Korea, PSA testing rates are not expected to reach 30% until around 2043, the lowest rate ever reported in the US. At that time, the number of new cases will be 43,864, nearly triple that of PCa patients registered in 2020 (16,815 cases).

Another benefit of this approach is that it allows the comparison of PCa incidence between the current trend and the chances of future public screening. As shown in Figure 2, general PSA testing could be suggested for a highly selected population, for example, only for those aged 60 and 70, to reduce unintended overdiagnosis and overtreatment. Given that even if a national screening program were implemented, it is unlikely that 100% screening with PSA would occur in all age groups and that screening rates would decline with age as compliance decreases, we limit the maximum value of the estimate in this hypothetical situation to 80%. If we postulated that 60% of Korean males aged 60 and 80% of those aged 70 years underwent PSA screening in 2030, the estimated number would be 37,503 in 2030. This figure is 30% higher when considering the original estimation following the current trend of a slight increase in PSA testing, which was 28,822. This anticipation suggests the possibility of a balanced PCa screening policy, focusing only on the most prevalent age groups. On the other hand, the current limited PSA screening rates may be maintained in younger populations. Given the relatively low-risk dominant disease profile in younger counterparts, active surveillance could be a practical option instead of active treatment. Repeated annual PSA screening may be ideal for detecting PCa, but for this study, we wanted to assume a minimum threshold that would reduce overdiagnosis while maintaining an effective detection rate. This requires data on the characteristics of the cancer, as the aggressiveness of PCa must be taken into account, but the current versions of KOSIS and NHIS data do not yet provide these characteristics, so the authors assumed public screening once every 10 years as a minimum standard.

The authors are aware of the limitations of this study. First, we did not utilize the authorized method, including an autoregressive integrated moving average model or time series analysis in future anticipation. However, even using the statistical approach of only projecting the data from the past, the number of estimated new cases did not match, as we summarized in Table 2. Given that PSA testing is the first chance to begin the suspicion of PCa in most clinical cases, an estimation not based on the incidence of PSA testing may not be practical. Second, as recent enhanced social interest in prostate disease developed in the elderly population, the rate of exposure to PSA testing has increased recently. Thus, our estimation based on the 2006–2016 period might be out of date. Therefore, PCa is predicted to be the most common type of cancer among men more quickly than our report suggests. Indeed, the number of articles anticipating the high incidence of PCa in many Asian populations keeps increasing. If PSA testing in each age subgroup affects the incidence of PCa, then an age-adjusted strategy considering the epidemic characteristics of PCa is imperative in establishing nationwide protection from PCa.

## 5. Conclusions

Due to the accelerating trend of social aging and the social awareness of prostate cancer continuing to grow, the incidence of patients with prostate cancer is projected to increase twofold by 2034 compared to 2020. If national screening using PSA tests were only carried out at the ages of 60 and 70, an almost threefold higher detection rate would be anticipated by 2040.

## Figures and Tables

**Figure 1 cancers-16-00503-f001:**
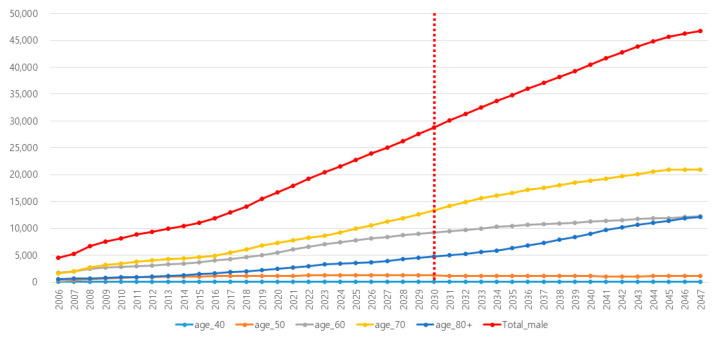
Estimated prostate cancer cases in Korea according to each age subgroup if the current trend of PSA testing incidence is maintained.

**Figure 2 cancers-16-00503-f002:**
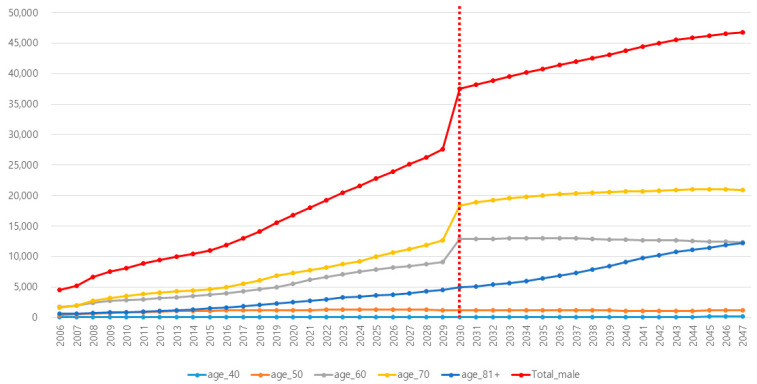
Estimated number of prostate cancer cases if PSA testing is included in the national screening program in 2030.

**Table 1 cancers-16-00503-t001:** Estimated number of new PCa cases and the anticipated PSA testing rate in each year (2021–2047).

	Reflected Incidence of PCa in Korea						Reflected Incidence of PSA Testing in Korea					
Year	Total	40s	50s	60s	70s	Over 80s	Total	40s	50s	60s	70s	Over 80s
2021	17,997	95	1235	6163	7767	2737	6.4	2.0	5.8	16.8	32.1	35.4
2022	19,227	97	1257	6606	8240	3026	7.1	2.1	5.9	18.4	34.1	38.2
2023	20,486	100	1262	7114	8713	3297	7.9	2.1	6.1	20.1	36.0	41.0
2024	21,606	102	1290	7496	9258	3461	8.7	2.2	6.2	21.7	38.0	43.8
2025	22,793	104	1290	7827	9937	3635	9.6	2.3	6.4	23.3	40.0	46.6
2026	23,921	106	1283	8167	10,632	3733	10.4	2.4	6.6	25.0	41.9	49.4
2027	25,108	109	1276	8462	11,256	4005	11.4	2.5	6.7	26.6	43.9	52.3
2028	26,310	111	1268	8727	11,925	4279	12.3	2.5	6.9	28.3	45.9	55.1
2029	27,563	113	1243	9050	12,625	4531	13.4	2.6	7.0	29.9	47.8	57.9
2030	28,822	115	1241	9283	13,383	4799	14.5	2.7	7.2	31.5	49.8	60.7
2031	30,076	118	1234	9507	14,237	4979	15.7	2.8	7.4	33.2	51.7	63.5
2032	31,317	120	1233	9792	14,876	5297	16.8	2.9	7.5	34.8	53.7	66.3
2033	32,543	122	1222	10,007	15,592	5600	18.0	2.9	7.7	36.5	55.7	69.1
2034	33,692	124	1202	10,309	16,143	5915	19.2	3.0	7.8	38.1	57.6	71.9
2035	34,871	125	1185	10,499	16,647	6415	20.5	3.1	8.0	39.7	59.6	74.7
2036	35,991	127	1177	10,659	17,157	6871	21.7	3.2	8.2	41.4	61.5	77.5
2037	37,090	129	1167	10,823	17,609	7362	23.0	3.3	8.3	43.0	63.5	80.0
2038	38,179	131	1155	10,978	18,029	7887	24.1	3.3	8.5	44.7	65.5	80.0
2039	39,300	133	1145	11,066	18,528	8429	25.2	3.4	8.6	46.3	67.4	80.0
2040	40,478	135	1125	11,250	18,917	9051	26.4	3.5	8.8	47.9	69.4	80.0
2041	41,679	137	1104	11,412	19,303	9723	27.6	3.6	9.0	49.6	71.4	80.0
2042	42,781	140	1096	11,597	19,754	10,195	28.8	3.7	9.1	51.2	73.3	80.0
2043	43,864	143	1103	11,744	20,122	10,753	30.0	3.7	9.3	52.9	75.3	80.0
2044	44,781	145	1120	11,851	20,587	11,078	31.1	3.8	9.4	54.5	77.2	80.0
2045	45,654	148	1146	11,967	20,910	11,483	32.1	3.9	9.6	56.1	79.2	80.0
2046	46,294	151	1167	12,119	21,002	11,854	32.9	4.0	9.8	57.8	80.0	80.0
2047	46,737	154	1185	12,265	20,963	12,171	33.5	4.1	9.9	59.4	80.0	80.0

**Table 2 cancers-16-00503-t002:** Estimated new PCa cases for 2022, 2030, and 2040 according to the literature.

Source	Publication Date	Data Source	Estimated Incidence in 2022	Estimated Incidence in 2030	Estimated Incidence in 2034	Estimated Incidence in 2040	Statistical Method	Variables Reflected
Jung et al. [21].	2022	National Cancer Incidence database (1999–2019)	22,391	-	-	-	Linear regression model	Age, period
Pak et al. [22].	2022	National Cancer Incidence database (1999–2016)	About 17,000	About 25,000	29,339	-	Age–period–cohort method	Age, period, cohort
International Agency for Research on Cancer (IARC) [23]	Last updated in December 2020	Global Cancer Observatory database (from National Cancer Registry, 2018)	13,873 (In 2020)	20,900	24,241 (In 2035)	26,828	Multiplying age-specific incidence	Age
Present study	2023	Korean Statistical Information Service Database (2006–2020)	19,227	28,822	33,692	40,478	Linear regression model	Age, period, PSA testing incidence (2006–2016)

## Data Availability

No new data were created.

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
