# Peer review of "The Short- and Long-Term Anticipation of Prostate Cancer Incidence in Korea: Based on Social Aging Trends and Prostate-Specific Antigen Testing Rate during the Last Decade"

_cancers, 2024, doi:10.3390/cancers16030503_

Round 1

Reviewer 1 Report

Comments and Suggestions for Authors

Comments:

    The manuscript describes " The Short and Long-term Anticipation of Prostate Cancer Incidence in Korea: Based on social-aging trends and prostate-specific antigen testing rate during the last Decade". This paper shows with the accelerated trend of social aging and the increasing social awareness of prostate cancer, the incidence of prostate cancer is expected to increase, but several points need to be clarified.

Comment:

1. Prostate cancer can be screened by testing the prostate-specific antigen (PSA) index. If the PSA index is greater than 10, the possibility of suffering from prostate cancer is as high as nearly 60%. The authors should describe prostate-specific antigen (PSA) diagnosis and analysis more in the introduction.

2. Prostate cancer can be screened through digital anal examination and blood testing for prostate-specific antigen (PSA). Common prostate cancer examinations also include medical history inquiry and prostate ultrasound. The authors should describe more of the differences between it and PSA in the introduction and discussion.

3. Where does the equation for PSA detection rate come from? The authors should explain.

4. Reasons for and solutions to the sudden increase in PSA testing rates that would be expected if public PSA screening were limited to men in their 60s and 70s (Figure 2). The author should be more descriptive in the Discussion.

Comments on the Quality of English Language

Minor editing of English language required

Author Response

Comments from Reviewer #1:

The manuscript describes " The Short and Long-term Anticipation of Prostate Cancer Incidence in Korea: Based on social-aging trends and prostate-specific antigen testing rate during the last Decade". This paper shows with the accelerated trend of social aging and the increasing social awareness of prostate cancer, the incidence of prostate cancer is expected to increase, but several points need to be clarified.

Comment:

Q1. Prostate cancer can be screened by testing the prostate-specific antigen (PSA) index. If the PSA index is greater than 10, the possibility of suffering from prostate cancer is as high as nearly 60%. The authors should describe prostate-specific antigen (PSA) diagnosis and analysis more in the introduction.

A1. We appreciate your kind comment. The role of PSA in prostate cancer is, as you point out, a key one. The original article was aimed at urologists, so we refrained from making basic comments. Still, given the interdisciplinary nature of Cancer, we have included a discussion in this revision of why PSA testing is important. In this revision, we have included a broader discussion of the following, including your points.

[Introduction session, Page 2] "In patients with no metastases and no symptoms, the detection rate of PCa is known to be directly related to PSA levels, which is not disease-specific but may raise suspicion of PCa. About 18% of patients with PSA in the 4 to 10 ng/mL range have a subsequent positive biopsy [8]. PSA levels greater than 10 ng/mL confer a greater than 67% likelihood of biopsy-detectable PCa [9]."

---------------------------------------

Q2. Prostate cancer can be screened through digital anal examination and blood testing for prostate-specific antigen (PSA). Common prostate cancer examinations also include medical history inquiry and prostate ultrasound. The authors should describe more of the differences between it and PSA in the introduction and Discussion.

A2. Thanks for the thoughtful point. While DRE tests play an important role alongside PSA in the detection of prostate cancer, unlike PSA, which is a numerical test, DRE tests are subject to the experience and subjective skills of the examiner. They are, therefore, less accurate in diagnosis. For this reason, unlike reports in the 90s, there has been an ongoing debate in recent years about the place of DRE in prostate cancer screening. Furthermore, each country has a different insurance system. In Korea, there is no separate insurance reimbursement for prostate cancer screening, making it somewhat difficult to conduct a national screening program. Prostate ultrasound is generally considered to have very low sensitivity for prostate cancer and has not been accepted for screening purposes. Recently, multiparametric MRI, which overcomes these shortcomings, has emerged as a standard imaging modality. Although large RCTs, including the PROBASE study, are ongoing, their cost-effectiveness as a routine screening test is still controversial. We've considered this in this revision and made the following new statements in the introduction and Discussion.

[Introduction session, Page 2] "Although digital rectal examination has been recommended for screening, its role is currently controversial [10], and multiparametric magnetic resonance imaging has been proposed to compensate for the lower sensitivity of ultrasonography. However, its validation for large-scale screening is still ongoing [11]."

 ---------------------------------------

Q3. Where does the equation for PSA detection rate come from? The authors should explain.

A3. This equation is the core of this thesis. As described in the main text, it is the statistical derivation of a formula to calculate the number of PCa cases each year based on the detection rate of PCa in Korea at each age in the past decade and the PSA screening rate at each age in the past decade. The creation of this regression model is already described in session 2, 'The statistics estimating the number of PCa population in the future' in Materials and Methods. By creating a formula with data from the past 10 years and applying it to age-specific data from the next 20 years, the result is the estimated future diagnosis rate of prostate cancer in Figure 1. Two professional statisticians were co-authors of the study to create this formula. Although data on future statistics based on the prevalence of prostate cancer in South Korea has been published for many years, the predicted data was much lower than the actual prevalence of prostate cancer detected, so the formula derived by the authors in this paper corrects the error of predicting with only conventional statistical methods by supplementing clinical patient outcomes. The differences in statistical methods and estimation results between our paper and previously published statistical estimation papers are already described in Table 2. In this revision, we've added the following to the Discussion for the benefit of our readers.

[Discussion session, page 5] "As such, the traditional statistical methods of population and period do not account for PSA screening rates, which are the primary means of detecting PCa in the clinical setting. However, this method makes it difficult to estimate accurately when PSA screening rates are low or change rapidly. With this in mind, the authors statistically derived an equation that can compensate for the fact that PSA screening rates and prostate cancer detection rates have varied dramatically by age in Korea over the past decade and used the results of this equation to estimate future projections."

 ---------------------------------------------------

Q4. Reasons for and solutions to the sudden increase in PSA testing rates that would be expected if public PSA screening were limited to men in their 60s and 70s (Figure 2). The author should be more descriptive in the Discussion.

A4. As previously described and shown in Materials and methods, the purpose of this study is to create a formula to estimate the number of future prostate cancer cases using data from the past 10 years, to compare it to the most recent data available, to compare it to the results of previously published prognostic studies, and to predict the number of prostate cancer cases over the next 20 years. Since the average incidence of prostate cancer in South Korea is highest in the 60s and 70s, as shown in Figure 1, we can define a minimum national mandatory PSA screening at these two ages. If this mandatory screening were to occur at these two ages only, the estimate at that time would be the revised future prostate cancer estimate shown in Figure 2. This point was not well articulated in the original article, and we have made it more explicit in this revision as follows.

[Materials and Methods sesseion, page 3] “We also calculated the anticipated PCa incidence if PSA testing were implemented as a national cancer screening program around 2030 only for men aged 60 and 70. This is a hypothetical situation, but assumes a collective implementation of minimal PSA screening in the two age groups with the highest incidence of PCa in the last decade of data.”

[Discussion session, page 7] “Given that even if a national screening program were implemented, it is unlikely that 100% screening with PSA would occur in all age groups and that screening rates would decline with age as compliance decreases, we limit the maximum value of the estimate in this hypothetical situation to 80%.”

Reviewer 2 Report

Comments and Suggestions for Authors

The manuscript presents an interesting idea for predicting the incidence of PCa but using an unauthorized method represents a major issue.

I would recommend the authors to use something with a more certified scientific impact.

Author Response

Comments from Reviewer #2:

Q1. The manuscript presents an interesting idea for predicting the incidence of PCa but using an unauthorized method represents a major issue. I would recommend the authors to use something with a more certified scientific impact.

A1. We appreciate your prudent remarks. This study did not use an unapproved method but rather statistically derived a new method that could compensate for the incorrectly predicted results of other studies using existing methods and estimated the future outlook based on the derived formula. There are already three papers published on the prediction of prostate cancer in Korea using traditional statistical methodologies based on existing age, duration, and population groups, and the methodologies used by them and their results are described in Table 2 in the text. However, despite using the same data accumulated over an almost similar period, the predicted values for the data in 2022 are quite different, as shown in the fourth column of Table 2. The authors recognize that the reason for this failure of the traditional methodology is the lack of clinical consideration of PSA, which is the main route by which prostate cancer is detected in the real world. This paper results from their attempt to compensate by disaggregating the traditional methodology, i.e., by using age-specific PSA screening rates. Two statisticians joined the paper as co-authors for such macro-scale statistical inference to add their expertise. Since the publication of this paper, the actual number of prostate cancer patients in 2021, the most recent year for which data are available, is 18,697, which we believe shows the reliability of this study as it is the most realistic among the studies published so far [Reference 1]. In this revision, we have made several changes to compensate for some reader confusion caused by the different approach from the original paper. Once again, we thank you for your review.

Reference 1. Updated Cancer Statistics in Korea [reported in December 29, 2023, National Cancer Cencer, Goyang, South Korea. Available from  https://ncc.re.kr/cancerStatsView.ncc?bbsnum=658&searchKey=total&searchValue=&pageNum=1

[Materials and Methods sesseion, page 3] “We also calculated the anticipated PCa incidence if PSA testing were implemented as a national cancer screening program around 2030 only for men aged 60 and 70. This is a hypothetical situation, but assumes a collective implementation of minimal PSA screening in the two age groups with the highest incidence of PCa in the last decade of data.”

[Discussion session, page 7] “Given that even if a national screening program were implemented, it is unlikely that 100% screening with PSA would occur in all age groups and that screening rates would decline with age as compliance decreases, we limit the maximum value of the estimate in this hypothetical situation to 80%.”

Round 2

Reviewer 1 Report

Comments and Suggestions for Authors

accepted

Comments on the Quality of English Language

Minor editing of English language required